# Exploring the Reported Strengths and Limitations of Aboriginal and Torres Strait Islander Health Research: A Narrative Review of Intervention Studies

**DOI:** 10.3390/ijerph20053993

**Published:** 2023-02-23

**Authors:** Romany McGuffog, Jamie Bryant, Kade Booth, Felicity Collis, Alex Brown, Jaquelyne T. Hughes, Catherine Chamberlain, Alexandra McGhie, Breanne Hobden, Michelle Kennedy

**Affiliations:** 1School of Medicine and Public Health, The University of Newcastle, Callaghan, NSW 2308, Australia; 2Hunter Medical Research Institute, The University of Newcastle, Callaghan, NSW 2308, Australia; 3Indigenous Genomics, Australia National University, Canberra, ACT 2601, Australia; 4Telethon Kids Institute, Nedlands, WA 6009, Australia; 5Rural and Remote Health, College of Medicine and Public Health, Flinders University, Darwin, NT 0810, Australia; 6Centre for Health Equity, School of Population and Global Health, The University of Melbourne, Parkville, VIC 3010, Australia; 7Judith Lumley Centre, School of Nursing and Midwifery, La Trobe University, Bundoora, VIC 3086, Australia; 8Health Behaviour Research Collaborative, School of Medicine and Public Health, College of Health, Medicine and Wellbeing, University of Newcastle, Callaghan, NSW 2308, Australia

**Keywords:** Aboriginal and Torres Strait Islander health, Indigenous health, review, intervention

## Abstract

High quality intervention research is needed to inform evidence-based practice and policy for Aboriginal and Torres Strait Islander communities. We searched for studies published from 2008–2020 in the PubMed database. A narrative review of intervention literature was conducted, where we identified researcher reported strengths and limitations of their research practice. A total of 240 studies met inclusion criteria which were categorised as evaluations, trials, pilot interventions or implementation studies. Reported strengths included community engagement and partnerships; sample qualities; Aboriginal and Torres Strait Islander involvement in research; culturally appropriate and safe research practice; capacity building efforts; providing resources or reducing costs for services and communities; understanding local culture and context; and appropriate timelines for completion. Reported limitations included difficulties achieving the target sample size; inadequate time; insufficient funding and resources; limited capacity of health workers and services; and inadequate community involvement and communication issues. This review highlights that community consultation and leadership coupled with appropriate time and funding, enables Aboriginal and Torres Strait Islander health intervention research to be conducted. These factors can enable effective intervention research, and consequently can help improve health and wellbeing outcomes for Aboriginal and Torres Strait Islander people.

## 1. Introduction

High quality research and evaluation that provides a sound evidence base can enable new understandings of health and disease, treatments and interventions, and directly inform the implementation of health policy that has measurable impacts on health outcomes [1]. Recently, the National Health and Medical Research Council (NHMRC) made commitments that at least 5% of the Medical Research Endowment Account would be spent on Aboriginal and Torres Strait Islander health and medical research [2]. The 2021 annual report card on Aboriginal and Torres Strait Islander health research funding shows this has been achieved, with 206 active grants funded in 2021 to the value of more than $58 million [3], exceeding the 5% target. However, more than 15 years on from the Australian Government’s ‘Closing the Gap’ strategy [4], the central commitment to close the life expectancy gap for Aboriginal and Torres Strait Islander peoples by 2031 was not on track [5]. As highlighted in the 2022 Lowitja Institute Close the Gap Campaign Report [6], to truly empower Aboriginal and Torres Strait Islander people and improve health and social outcomes, it is critical that ongoing investment is embedded in research that will inform policy and practice reform.

Many distinct types of research provide differing and complementary insights into complex health problems. Interventions and evaluations have the capacity to directly impact health outcomes and policy through the translation of evidence into practice [7]. While randomised controlled trials have historically represented the gold standard in determining effectiveness [8], evaluation and implementation trials facilitate the integration of evidence-based interventions into health policy and practice [7]. Interventions and evaluations present significant challenges to conduct [9], as they are often costly, time-intensive and logistically complex, requiring the involvement of multiple sites and stakeholders and/or large sample sizes. As a result, there has been limited intervention research conducted in Aboriginal and Torres Strait Islander health [10,11].

Critically, Indigenous people globally have reported negative experiences and impacts of research [12]. Historically, research undertaken with Aboriginal and Torres Strait Islander people has been conducted largely with Aboriginal and Torres Strait Islander people as research subjects, with no control, ownership, or conceptual involvement [13,14,15]. Consequently, understanding strengths and limitations of intervention and evaluation research conducted with Aboriginal and Torres Strait Islander people can provide guidance for future researchers to achieve optimal outcomes for the communities they serve [16], and identify focus for ensuring research and research funding ultimately benefits health outcomes. This narrative review aims to understand the strengths and limitations, as reported by researchers, of conducting Aboriginal and Torres Strait Islander health research interventions and evaluations.

## 2. Materials and Methods

### 2.1. Literature Search

The current review is a secondary analysis of a subset of studies from a larger review [11]. The parent review involved a systematic search of the literature via the Lowitja Institute website using the Lowitja.search tool to identify original research in the PubMed database focused on Aboriginal and Torres Strait Islander health in Australia. Searches were restricted to studies published from 2008 (i.e., since the introduction of the ‘Close the Gap’ campaign) to December 2020.

### 2.2. Inclusion and Exclusion Criteria

Studies were included if they: (i) reported original research; (ii) were focused on Aboriginal and Torres Strait Islander health in Australia; and (iii) used intervention designs, including evaluations, trials (randomised and non-randomised), pilots, and implementation studies. Descriptive research studies (e.g., case studies, cross-sectional, cohort or qualitative research) and non-data-based studies (e.g., protocols, reviews and opinion pieces) were excluded.

### 2.3. Data Extraction and Synthesis

Identified studies were classified by research type by one of three authors (MK, KB, and RM). Location and remoteness were extracted for all studies. For each study meeting eligibility criteria, data was extracted using NVivo by RM (non-Indigenous), MK (Wiradjuri), and FC (Gomeroi) to identify researcher reported strengths and limitations of their research practice. Strengths and limitations reported in studies applicable to both conducting research itself and research impact were extracted, as impacts on the research process ultimately impact research outcomes. FC cross-checked 20% of the studies, and no discrepancies were found. A narrative review was selected to synthesise the results as this is an appropriate method for summarising strengths and limitations to the research process [17]. The strengths and limitations as detailed below are from the perspective of researcher teams, rather than the Aboriginal and Torres Strait Islander communities involved in the research, as this is how they are reported in publications.

## 3. Results

### 3.1. Search Results

A total of 2150 studies reporting original research and examining Aboriginal and Torres Strait Islander health research were identified. From these, we extracted studies which reported using an evaluation (*n* = 110), pilot intervention (*n* = 30), trial (*n* = 56) or implementation (*n* = 44) design, which resulted in 240 studies that were included in the analysis (see Figure 1).

### 3.2. Frequency of Reported Strengths and Limitations

In the Appendix A provides the numbers and percentages of the reported strengths and limitations by publication classification. For the sake of brevity, we report the number of references rather than outlining the specific references. For a full summary of each reference and their corresponding publication classification, journal, state, remoteness, and strengths and limitations, please see the Appendix A.

### 3.3. Reported Strengths

#### 3.3.1. Community Engagement and Partnerships


*“Working partnerships were essential for achieving the program’s positive outcomes. This requires the meaningful engagement with a variety of partners at each location, including other service providers, local Aboriginal organisations, Elders and Traditional Owners and public authorities”*
—Blignault et al., 2016 [18] (Evaluation)

Community involvement, engagement, consultation, and collaboration were reported as key strengths in 64 studies. Community consultation and collaboration were inclusive of engagement with Elders and Traditional Owners, Aboriginal health services, and other Aboriginal and Torres Strait Islander organisations and were reported across different phases of the research process. This was in the form of partnerships, co-design, and interpretation of findings. These were identified as important in ensuring that programs meet community identified priority areas. Community involvement was often recognised as an ongoing/continuous relationship, constituting a partnership rather than one off consultation, which was consistently considered a benefit to the research process.

#### 3.3.2. Sample Qualities


*“Strengths of this study were that the study population was drawn from five communities spread across the NT that were randomly selected, thus giving confidence of the generalisability of the results to other like, remote NT communities”*
—Brimblecombe et al., 2018 [19] (Trial)

Thirty-two studies identified properties of the sample as a strength. This included large sample size, a diverse sample, and/or generalisable sample. A large sample size was considered to give confidence of comprehensive results and provide rigour, while a diverse sample was considered to contribute to the broad representation of participants and geographical reach of the programs. Most of the studies that spoke of diversity as a strength predominantly referred to this in terms of geographic location or ability to reach remote settings, rather than in relation to age or gender.

#### 3.3.3. Aboriginal and Torres Strait Islander Research Team


*“The Aboriginal leadership of the project was crucial in enabling the VACCHO nutrition team to build the capacity of both Aboriginal community organisations and mainstream organisations across the sector”*
—Genat et al., 2016 [20] (Evaluation)

Twenty-one studies recognised having Aboriginal and Torres Strait Islander people as part of the research team as a strength. Some noted Aboriginal leadership as key for meaningful engagement, ensuring community priority areas were met, delivering a culturally safe environment, building rapport, and enabling the research team to contribute to capacity building of communities and Aboriginal and Torres Strait Islander community organisations. Within these studies, some specifically referred to Aboriginal and/or community leadership over the whole program. Additionally, some studies highlighted the importance of Aboriginal governance in their project in decision making and guidance processes. Similarly, in several studies, Aboriginal research staff were considered important, such as helping with recruitment, program uptake, and retention rates.

#### 3.3.4. Culturally Appropriate and Safe Research Practice


*“On the whole, the removal of cost barriers and the creation of welcoming, culturally safe spaces appeared to make the greatest contribution to increased access to chronic illness prevention and management services by Indigenous people”*
—Bailie et al., 2015 [21] (Evaluation)

Seventeen studies reported the use of culturally appropriate and safe research practices as key strengths of their research. Programs that were designed to be run by community members contributed to ensuring cultural appropriateness. Having a culturally appropriate study, space, and health information was reported as critical to engaging local community, although few details of what this entailed were provided. The few studies that did provide details outlined that culturally safe environments in research often were the result of community-led, implemented, and supported research. Some studies highlighted the benefit of using methods well-suited for Indigenous research projects, such as yarning, as a useful way to help strengthen relationships and support between researchers, participants, and community.

#### 3.3.5. Capacity Building Efforts


*“The predominant strength of this study was the involvement of Aboriginal research staff within the two participating health services and the associated follow up. This provided a culturally appropriate approach to data collection, capacity building for Aboriginal and Torres Strait Islander staff, students and health service providers, and identified factors that would need to be considered in future studies”*
—O’Grady et al., 2015 [22] (Pilot intervention)

Capacity building efforts were reported as a strength in twelve studies. This included training, employment, knowledge building, and program delivery which was frequently linked to positive outcomes such as adherence to cultural practices and uptake of resources. Ten studies detailed the way that their program provided support or training to professional roles, such as Aboriginal Health Workers (AHWs) and service providers. Several studies commented on the benefit of capacity building, including ensuring culturally appropriate practice, better engagement with the study by participants, and good recruitment rates.

#### 3.3.6. Providing Resources and Reducing Costs for Services and Communities


*“The success of the program was the result of identifying the key issues, involving key experts in the development of the materials, and creating a supportive and robust infrastructure within the health care organisation by making the resources available for future use by staff”*
—Khalil et al., 2019 [23] (Implementation)

Providing resources and reducing costs were reported as an enabler for research in nine studies. One area that was highlighted in many studies was the benefit of training AHWs within their programs which offered opportunities for professional development at lower costs. This training can have a flow-on effect to reduce the need and cost of having external professionals visit services. Additionally, many studies highlighted that providing funding or resources for services meant that these services could better provide for their communities’ healthcare needs.

#### 3.3.7. Understanding Local Culture and Context


*“The process of implementing the EACHS policy into service delivery identified the importance of understanding local populations, service provision, and the cultural aspects of care. This local knowledge was also required to encourage agencies to use a collaborative approach when implementing the EACHS policy”*
—Bradshaw et al., 2015 [24] (Implementation)

Understanding and incorporating local culture and context was considered a key strength in seven studies by ensuring meaningful and targeted interaction with community and optimising engagement through the implemented programs. Additionally, the ability to adapt the program to suit priority areas as identified by local community was important to study success.

#### 3.3.8. Appropriate Timelines for Completion


*“The five year time frame enabled the research team to build relationships of trust with community members and service providers, demonstrate preparedness to act on community recommendations, and have a sufficiently sustained presence to make the most of opportunities that presented themselves”*
—Robertson et al., 2013 [25] (Implementation)

Two studies noted the benefits of having what they deemed to be an appropriate timeframe for completion of the studies. This strengthened the ability to build relationships and appropriately respond to community recommendations. Longer time frames also provided an opportunity to capture costs over multiple years.

### 3.4. Reported Limitations

#### 3.4.1. Difficulties Achieving the Target Sample Size


*“Our difficulties in achieving recruitment targets were multifactorial, and barriers occurred at both institutional and individual participant levels”*
—Peiris et al., 2019b [26] (Trial)

The most frequently reported limitation identified across studies related to difficulties achieving the target sample size, which was reported in 119 studies. This was often noted as having an impact on the generalisability of the research and contributed to lack of statistical power, particularly in randomised controlled trials. Difficulties achieving the target sample size were attributed to staff turnover, reported difficulties with participant recruitment and difficulties in engaging Aboriginal and Torres Strait Islander communities. Additionally, a number of studies reported that high attrition rates limited their findings. Some studies indicated that attrition impacted statistical power, increased potential bias, reduced researchers’ ability to interpret findings, and created challenges for participant follow-up.

#### 3.4.2. Inadequate Time to Conduct Research


*“The project’s timeframe further limited the potential sample size. Funding was allocated in September 2016, recruitment commenced in November 2016, and due to the non-negotiable end date of 30 June 2018, recruitment ended in December 2017. A longer recruitment period would likely have increased participation”*
—Askew et al., 2019 [27] (Evaluation)

Inadequate time was the second most frequently reported limitation to conducting Aboriginal and Torres Strait Islander health intervention research, which was reported in 23 studies. Across the included publications, time restrictions appeared to be an overarching barrier to various components of research, such as the inability to monitor the ongoing effectiveness of interventions over time, reduced recruitment ability, reduced training capacities, impact on overall rigour, and not enough time for proper community engagement and education. The process of gaining ethical and governance approvals were reported as lengthy and interrupted the research process, contributed significantly to time pressures, and prolonged recruitment. This included delays in gaining approval and the total time required for the ethics review process. Requiring approvals from various governance groups and ethics committees can also lengthen the time required to set up a project.

#### 3.4.3. Funding, Cost and Resources


*“The main limitation of the evaluation was the lack of time and funding to enable more extensive community feedback and verification of the findings”*
—Lowell et al., 2015 [28] (Evaluation)

Insufficient funding was identified as a limitation in 15 studies. Ongoing relationships with communities required for quality research was inhibited by time and funding limitations. This also clearly impacted the ability to disseminate results efficiently back to the communities. Other studies identified that insufficient funding limited the overall quality of research as it only allowed for the completion of small-scale projects. Limited funding timeframes were reported to also impact recruitment ability. Additionally, limited access to required resources, such as an appropriate workspace or limited transport options impeded the effective implementation of research.

#### 3.4.4. Limited Capacity of Health Workers and Services


*“Some AHWs were clearly ‘stretched’ by multiple demands and unable to find time in their busy schedules”*
—McRae et al., 2008 [29] (Evaluation)

The limited capacity of health workers/professionals and services involved was reported as a limitation in 12 studies. Researchers reported that limited staff capacities, particularly among AHWs, resulted in difficulties undertaking data collection, conducting interviews with health professionals, and engaging staff in implementing programs. This was often attributed to understaffing and heavy demands on AHWs.

#### 3.4.5. Lack of Community Involvement and Communication


*“We acknowledge the greatest reason for the study challenges was likely that, although addressing a priority of kidney health and dental stakeholders, it did not necessarily address an identified priority of Aboriginal communities and ACCHOs”*
—Jamieson et al., 2020 [30] (Implementation)

Three studies reported lack of community involvement in the research process as a considerable limitation. From the limited details provided, it was reported that community involvement was limited but was crucial for future research, there were challenges with obtaining community input, and acknowledging the importance of Aboriginal leadership and involvement in clinical trials. Three studies reported barriers in communication between the researchers and communities, which related to the importance of making research inclusive for different language groups.

## 4. Discussion

This review offers insight into researcher reported strengths and limitations of conducting Aboriginal and Torres Strait Islander health intervention research. Lack of time and funding were major barriers that often coincided with other reported limitations including difficulties achieving sample size, monitoring ongoing impact of the intervention, ability to create and maintain meaningful connection with the community, training and education development, and difficulty obtaining ethical and institutional approvals. Similarly, studies discussed the importance of meaningful, ongoing relationships and involvement from community in producing quality research. Often, this negated many of the reported barriers and turned them into strengths of the research process, such as the ability to produce larger, representative sample sizes and capacity building through training and education. These findings highlight the impact that proper consultation, relations and leadership from community coupled with appropriate timeframes and funding, can have on the conduct of quality research and by extension, better health and wellbeing outcomes for Aboriginal and Torres Strait Islander people. This is consistent with other research, which demonstrates that short-term funding contributes to service delivery issues, while additional resources enhanced outcomes and future funding [31].

To provide precise estimates of treatment effects, studies evaluating the effectiveness of new interventions require an adequate sample size. Aboriginal and Torres Strait Islander people currently make up only 3% of Australia’s population and are spread across diverse locations around Australia [32]. Consequently, as with other small populations, recruitment can be particularly challenging and often require different sampling techniques to recruit participants [33]. Therefore, this highlights the need for realistic sample size calculations to be performed early in planning an intervention, and the consideration of alternative study designs.

Furthermore, in line with ethical guidelines [34,35,36] Aboriginal and Torres Strait Islander people should be actively and appropriately involved in all aspects of research about them, including the design and conduct of the research, ownership of data, interpretation of data, and the reporting and publication of findings. Research has shown that Indigenous communities will support health interventions if it reflects their cultural beliefs and enhances conditions for community determined by community [37]. This is also critical if investments in Aboriginal and Torres Strait Islander health research are going to reduce the burden of disease and increase the likelihood of positive health and wellbeing impacts [38]. Our results showed that research was strengthened by Aboriginal and Torres Strait Islander involvement in terms of leadership, research team members, and the wider community. Research should therefore be conducted with and by Aboriginal and Torres Strait Islander communities instead of on them [15].

Developing, implementing and disseminating research with Aboriginal and Torres Strait Islander people ethically requires longer time frames to establish partnerships between researchers and community [39] and to obtain appropriate ethical approvals [35,40]. Sufficient time is needed ensure that researchers can enable true partnership, and obtain the required governance, ethical and institutional approvals, without impacting their ability to complete research within the funding period. This may mean rethinking current arrangements whereby grant funding is not released until ethical approval is given. Grant funding should include increased funding for resources (such as training programs, educational material, Aboriginal and Torres Strait Islander personnel, and research support staff) and offer flexible timeframes to acknowledge community timelines and ethical engagement necessary for appropriate implementation. Funding is also needed for knowledge translation activities to ensure Aboriginal and Torres Strait Islander communities are informed about research outcomes and findings. Research in Canada has demonstrated the importance of localised Indigenous ways of knowing, dissemination and involvement in effective knowledge translation, to inform health policy and future research [41,42], which therefore warrant appropriate funding.

Researchers must acknowledge the demands and priorities on Aboriginal medical services and their staff, and that Aboriginal medical services are not funded and resourced to conduct research [43,44]. Researchers and funding bodies must ensure that research is not a financial burden on health services and communities, and that appropriate human and other resources are provided. There is currently limited evidence on the interest and capacity of Aboriginal Community Controlled Health Services to engage in and lead research.

It is important to highlight that some of the strengths and limitations identified in this review are not necessarily unique to Aboriginal and Torres Strait Islander health research. For example, strengths and limitations related to sample size are commonly reported in various types of research outside of Indigenous-specific research [45]. However, previous research has identified unique barriers and enablers to Aboriginal and Torres Strait Islander research [46,47,48]. The results from our review highlight that strengths such as Aboriginal leadership and community engagement can enable research in Aboriginal and Torres Strait Islander health, and also help mitigate barriers such as sample size. Identifying common strengths and limitations can inform future researchers on how to optimise the design, partnership and conduct of intervention research with Aboriginal and Torres Strait Islander communities.

## 5. Conclusions

Findings from this review indicate that intervention research in Aboriginal and Torres Strait Islander health is enabled when it is led by Aboriginal and Torres Strait Islander people and embeds Aboriginal and Torres Strait Islander researchers and community across all phases of the research. Funding applications for research projects in this field need to include appropriate funding and resourcing that is allocated to these best practice approaches to allow meaningful and impactful implementation of Aboriginal and Torres Strait Islander health research (for example, the new changes to NHMRC grants from 3 years to 5 years funding). Longer and more flexible funding timeframes for large scale interventions and evaluations allows for respect of Aboriginal and Torres Strait Islander community timelines and timeframes required for appropriate and ethical community engagement. This is particularly important when considering the complexities of multi-state ethics processes. Shared learning and reflective practice are one approach researchers can take to do better when it comes to Aboriginal and Torres Strait Islander health research.

## Figures and Tables

**Figure 1 ijerph-20-03993-f001:**
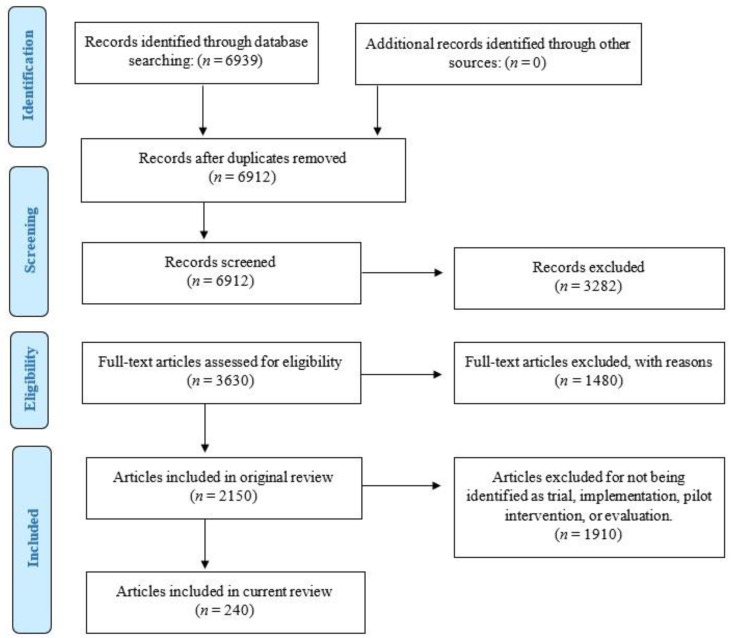
Preferred Reporting Items for Systematic Reviews and Meta-Analysis four-phase flow diagram.

## Data Availability

The articles included in this review are referenced in the Appendix A.

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
