# Peer review of "Exploring the Reported Strengths and Limitations of Aboriginal and Torres Strait Islander Health Research: A Narrative Review of Intervention Studies"

_ijerph, 2023, doi:10.3390/ijerph20053993_

Round 1

Reviewer 1 Report

Please see attached review report.

Author Response

  1. Abstract: Lines 28-30: limitations are reported here first. It should be the other way around – reported strengths should come first on these lines followed by reported limitations.

We have adjusted the order of these lines in the Abstract.

  1. Introduction: Line 45: Define NHMRC for the international audience in this first instance – National Health and Medical Research Council

We have provided the full title on Line 45.

  1. Results: Table 1: The presentation of n in Table 1 is a bit confusing, i.e. the columns of Evaluations (n =110), Implementations (n = 44), Pilot interventions (n = 30), and Trials (n = 56) have reported n, and for Strengths and Limitations have their n each. Lines 120-122 should clarify this better.

To increase the clarity of the manuscript, we have now moved this table to supplementary material. Within the table, the quotes column has been removed and the quotes incorporated into the beginning of each theme in the main manuscript. To further clarify the table, we have removed the numbers from publication classification row, and included a note at the bottom of the table:

“Note. Evaluations (n =110), implementations (n = 44), pilot interventions (n = 30), and trials (n = 56). The cells represent the number and percentages of studies within the theme by publication classification.”

  1. Results: Line 171: Add proportion (%) to ‘twelve studies’.
  2. Results: Section 3.5: The number of studies reported in the Reported Strengths section 3.4 includes %, but the number of studies reported in the Reported Limitations section 3.5 does not include %. For consistency either use % in both sections or not at all.

To address Points 4 and 5, we have focused on numbers rather than percentages, and subsequently removed percentages from the text.

  1. Discussion: Line 267: Delete ‘based’ after ‘performed’ in the sentence.

This word has been deleted.

  1. Discussion: Line 311-314: Please revise the sentence to specify which unique barriers and enablers, and which strengths and limitations are intended to be highlighted here. It’s not clear at the moment.

The last paragraph of the Discussion has been edited to provide clarity about specific strengths which can enable research and help mitigate barriers:

“For example, strengths and limitations related to sample size are commonly reported in various types of research outside of Indigenous-specific research.[29] However, previous research has identified unique barriers and enablers to Aboriginal and Torres Strait Is-lander research.[30-32] The results from our review highlight that strengths such as Ab-original leadership and community engagement can enable research in Aboriginal and Torres Strait Islander health, and also help mitigate barriers such as sample size. Identifying common strengths and limitations can inform future researchers on how to optimise the design, partnership and conduct of intervention research with Aboriginal and Torres Strait Islander communities.”

Reviewer 2 Report

Table 1. Summary of the strengths and limitations reported in the studies. Need to be concisely described rather than elaborating and misfitting tables.  The reason for selecting n=240 needs to be well described.

Author Response

Based on feedback from Reviewer 1, we have revised this table and moved it to Supplementary material. Within the table, the quotes column has been removed and the quotes incorporated into the beginning of each theme in the main manuscript. We have also edited the section 3.1. Search Results to specify that we extracted articles that were evaluation, pilots, trials, or implementations, which resulted in 240 studies:

“A total of 2,150 studies reporting original research and examining Aboriginal and Torres Strait Islander health research were identified. From this, we extracted studies which reported used an evaluation (n = 110), pilot intervention (n = 30), trial (n = 56) or implementation (n = 44) design, which resulted in 240 studies that were included in the analysis (see Figure 1).”

Reviewer 3 Report

This paper is on an important topic but it read as a list of documents, with statistics about the themes raised but what was said  about the actual themes themselves was superficial. Table up front was way too long and hard to read. Further, it was really presented as a systematic not a narrative review as suggested in the paper. The discussion did not add to the overall paper and while what is did state was very strong, it wasn't new  - this is standard and ethical practice for engaging with ATSI communities - what does this paper add to that? There is further, especially for an international audience, no explanation of the particular characteristics of ATSI communities. Overall, I do not think the paper really helped build understanding into what is an important topic. Given the fact that there were 240 docs analysed, this was a difficult and disappointing paper that would need significant reworking to make it a valuable contribution - authors need to actually do a narrative review and give much deeper insight into what key findings were and why they matter in this cross cultural context.

Author Response

  1. Table up front was way too long and hard to read.

Based on feedback from other reviewers, we have moved the table to the supplementary material and removed the quotes.

  1. Further, it was really presented as a systematic not a narrative review as suggested in the paper. The discussion did not add to the overall paper and while what is did state was very strong, it wasn't new - this is standard and ethical practice for engaging with ATSI communities - what does this paper add to that? There is further, especially for an international audience, no explanation of the particular characteristics of ATSI communities. Overall, I do not think the paper really helped build understanding into what is an important topic. Given the fact that there were 240 docs analysed, this was a difficult and disappointing paper that would need significant reworking to make it a valuable contribution - authors need to actually do a narrative review and give much deeper insight into what key findings were and why they matter in this cross cultural context.

To strengthen the structure of the paper in terms of a narrative review, we have placed a key quote at the beginning of each theme, removed the overuse of numbers/percentages from the text, and added additional content to the introduction and discussion. The data came from a larger review which was systematic, and we felt it was important to be transparent about this. However, as stated in section 2.3, a narrative review approach was used as it was more appropriate for analysing reported strengths and limitations rather than results of the research.